# Numerical Study of Growth Rate and Purge Time in the AlN Pulsed MOCVD Process

**Wei-Jie Lin and Jyh-Chen Chen ***

Department of Mechanical Engineering, National Central University, Zhongli 320, Taiwan
* Correspondence: jcchen@ncu.edu.tw

**Abstract:** The relationship between the purge time and the overall growth rate in pulsed injection metal–organic chemical vapor deposition with different V/III ratios is studied by numerical analysis. The transport behavior of TMAl and TMAlNH$_3$ during the process is studied to understand the effect of the adductive reaction. The results show that, as the V/III ratio increases, there is a significant reduction in the average growth rate per cycle, without the addition of a purging H$_2$ pulse between the III and V pulses, due to stronger adductive reaction. The adductive reaction can be reduced by inserting a purging pulse of pure H$_2$ between the III and V pulses, but there is a decrease in the overall growth rate due to the longer cycle time. At smaller V/III ratios, the growth rate decreases with increasing purge times, since the gain in reducing the adductive reaction is offset by the detrimental effect of extending the cycle time. The degree of reduction in the adductive reaction is higher for larger V/III ratios. When the benefit of reducing the adductive reaction overcomes the deficiency of the extending cycle time, a remarkable enhancement of the growth rate can be obtained at high V/III ratios by inserting a pure H$_2$ purge pulse between the III and V pulses. A higher overall growth rate can be achieved at higher V/III ratios by choosing an appropriate purge time.

**Keywords:** MOCVD; nitride; pulsed injection; growth rate

## 1. Introduction

III-nitride materials are widely used in power devices, high-frequency communications, and UV applications because of their excellent properties. Among the many III-nitride materials, aluminum nitride (AlN) is one of the most important, often used as a buffer layer between the sapphire substrate and other III semiconductor materials, because the lattice mismatch between AlN and sapphire is minor. However, the small migration of Al atoms on the substrate makes it difficult to grow high-quality AlN film [1]. One way to increase the migration of the Al atoms is to increase the temperature of the substrate. However, the high-temperature growth conditions not only make the parasitic reaction more intense, they also induce the problem of thermal expansion. Another way to increase the migration of Al atoms is to apply the pulsed injection method (PI method) during the metal–organic chemical vapor deposition (MOCVD) process. During the pulsed process, the supply of TMAl and NH3 is carried out separately. The TMAl follows the pyrolysis reaction pathway instead of adducting with NH$_3$ in the gas phase before being deposited on the surface. It changes the dominant species on the surface from Al-N to Al. The migration of Al atoms increases because the number of N atoms is much lower during the pulsed process [2]. Smoother AlN film can be achieved due to the lateral growth of film caused by the higher Al migration. Demir et al. [3] grew high-quality AlN by pulsed MOCVD (PIMOCVD) at a growth temperature of 1443 K with III and V pulse durations of 4 s and 2 s, respectively.

Many studies have pointed out the need for a hydrogen purge between the TMAl pulse (III pulse) and NH$_3$ pulse (V pulse) to further suppress the adductive reactions and achieve better surface roughness. Khan et al. [4] fixed the duration of the different pulses to 1 s with a 1 s hydrogen purge time in between. The crystalline quality of the material

grown by the pulsed process at 723 K was as good as the material grown by the steady-flow process at 1273 K. Takeuchi et al. [5] compared the growth rates of AlN film grown by the steady-flow process and pulsed process with a 1 s purge time. In the steady-flow process, the TMAl and NH3 were supplied at the same time. At a V/III ratio of 20.8, it was found that the growth rate of the pulsed process was 5.56 times that of the steady-flow process.

The effects caused by different TMAl and NH3 flow rates have also been investigated. Kröncke et al. [6] varied the effective V/III ratio between 200 and 700 and found that increasing the purge time after supplying TMAl could reduce the density of pits. The growth rate of the AlN film for these processes was about 1 $\mu$m/h. Rahman et al. [7,8] grew AlN films using $NH_3$ flow rates ranging from 0.2 to 2.2 slm, with a fixed TMAl flow rate of 15 slm. The durations of the III and V pulses were 4 s and 2 s, respectively, and the growth temperature was 1453 K. The results showed that the best AlN film quality was obtained with 0.6 slm $NH_3$. The growth rate of the AlN films was between 0.225 and 0.249 $\mu$m/h.

The potential of the pulsed method to improve the quality of AlN has been confirmed in numerous studies. However, there has been great divergence in the growth conditions used in different works. The pulsed process includes more external control parameters than a steady-flow process, making the transport phenomenon during the growth process more complicated. In order to obtain optimal growth conditions, it is important to have an in-depth understanding of its transport phenomena. The transport phenomena during the pulsed process have been investigated numerically in several works. Nakamura et al. [9] performed numerical experiments and observed the generation of AlN particles with different purge times (0, 1, and 2 s) and a constant V/III ratio of about 4000. The results showed the mole concentration of AlN particles to be eliminated with a purge time of 2 s. However, there was a decrease in the overall growth rate per cycle as the purge time increased. Endres et al. [10] also carried out numerical experiments. They showed that for a V/III ratio of 20.8, the optimal purge duration would be 5–6 times the duration of the III and V pulses, resulting in the highest growth rate. Lin and Chen [11] conducted a numerical investigation to explore the physical mechanisms in the evolution of flow motion and heat and mass transfer during the pulse process. They found that the duration of the III and V pulses does not affect the particle generation if the duration time is longer than a critical value (a pulse steady state is reached). The diffusive mass transport of $NH_3$ in $H_2$ is greater than that of TMAl in $H_2$ due to the higher mass diffusion coefficient of $NH_3$ in $H_2$. Therefore, the V pulses easily exceed the III pulses before reaching the substrate when the III/V ratio is 1. More AlN particles occur in the high-temperature region during the $H_2$ pulse after the III pulse. To reduce the occurrence of AlN particles, the duration of the $H_2$ pulse after the III pulse should be longer than that after the V pulse. However, the relationship between the purge time and the V/III ratio during the pulsed process and the transport phenomena is still unclear. When the V/III ratio is increased, the purge time should be extended, since more $NH_3$ will remain in the chamber, leading to stronger parasitic reactions. Furthermore, at a higher V/III, the transient mass transport process becomes more complex, since the main species of the V pulse changes from $H_2$ to $NH_3$.

In this study, the numerical scheme developed in our previous study [11] is extended to first study the transport phenomenon and overall growth rate of the cases without purging and with different V/III ratios. Then, the relationship between the purge time, V/III ratio and overall growth rate is investigated and the maximum overall growth rates for different V/III ratios are investigated.

## 2. Modeling

Figure 1 shows a schematic diagram of the simplified horizontal MOCVD reactor built in our previous study [11] based on the study of Chen et al. [12]. The temperature of the substrate surface was maintained at 1200 K and the operating pressure was 85 Torr. For pulsed processes, the total flow rate for all pulse sequences was fixed at 7 slm to maintain the stability of the flow field. The flow rates of the III and V pulses are denoted by $Q_{TMAl,p}$ and $Q_{NH3,p}$, respectively. The molar flow rate of TMAl for the III pulse was kept

at 30 µmol/min. The V/III ratio was varied from 1 to 10,000. In the previous study, the concentration of precursors was found to remain stable at 0.15 s after the start of the III or V pulse, which is close to the effective residence time (about 0.13 s). Therefore, in this study, the supply time of the TMAl and NH3 was fixed at 0.15 s and the pure hydrogen purge time between III and V pulses was changed from 0 s to 0.1 s. The working fluid in the chamber was considered as an ideal gas. The Reynolds number for the cases investigated in this study was below 400. Compressible laminar flow was applied to describe the flow motion during the process. The equations for the conservation of mass, momentum, heat energy, species, and the detail of material properties are listed in the previous work [10]. The reaction model follows the work performed by Mihopoulos [13]. The commercial software COMSOL was applied to solve the MOCVD processes coupling the fluid dynamics, the heat and mass transfer, and the chemical reactions by the finite element method. The discretization for the momentum and heat equation was linear and that for mass transport and reaction model was quadratic. Due to the rapid chemical reaction near the inlet, the distribution of the width of the element was arranged as arithmetic distribution and the size of the smallest element was 1 µm. The distribution of the height of the elements near the chamber wall also followed arithmetic distribution to calculate the boundary layer effects and the mass flux caused by surface reactions. The backward differentiation formula (BDF) was used to solve the transient behavior and the maximum order for the BDF was 2. To investigate the transport behavior of different species during the process, the maximum time step was limited by 0.001 s. The simulation results obtained in the steady-flow process with this scheme in our previous study [11] are in good agreement with previous results [10,12,13].

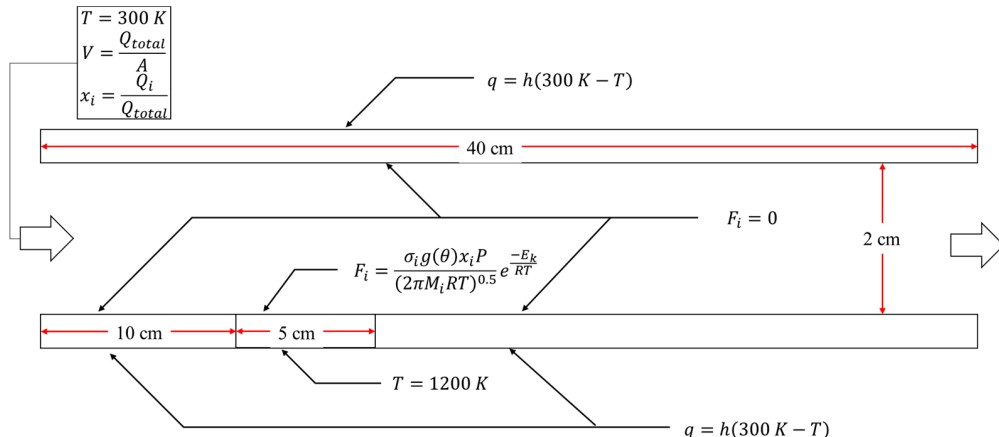

**Figure 1.** Schematic diagram and boundary conditions of a simplified horizontal MOCVD reactor.

### 3. Results and Discussion

Before studying the pulsed process, it is necessary to understand the effect of the V/III ratios during the steady-flow process. Figure 2 shows the relationship between the average growth rate along the entire wafer surface and the V/III ratio when the operating pressure is 40 Torr. As the V/III ratio increases, the growth rate decreases due to the higher generation of Al particles caused by higher amounts of $NH_3$. This trend is similar to the numerical and experimental results presented in [14] for a close-coupled showerhead (CCS) reactor during the steady-flow process. However, the growth rate obtained in this study is higher than in the CCS results. This may be due to the difference in growth area between the CCS reactor used in ref. [14] ($60.8 \text{ cm}^2$) and the channel reactor ($25 \text{ cm}^2$) used for the present study.

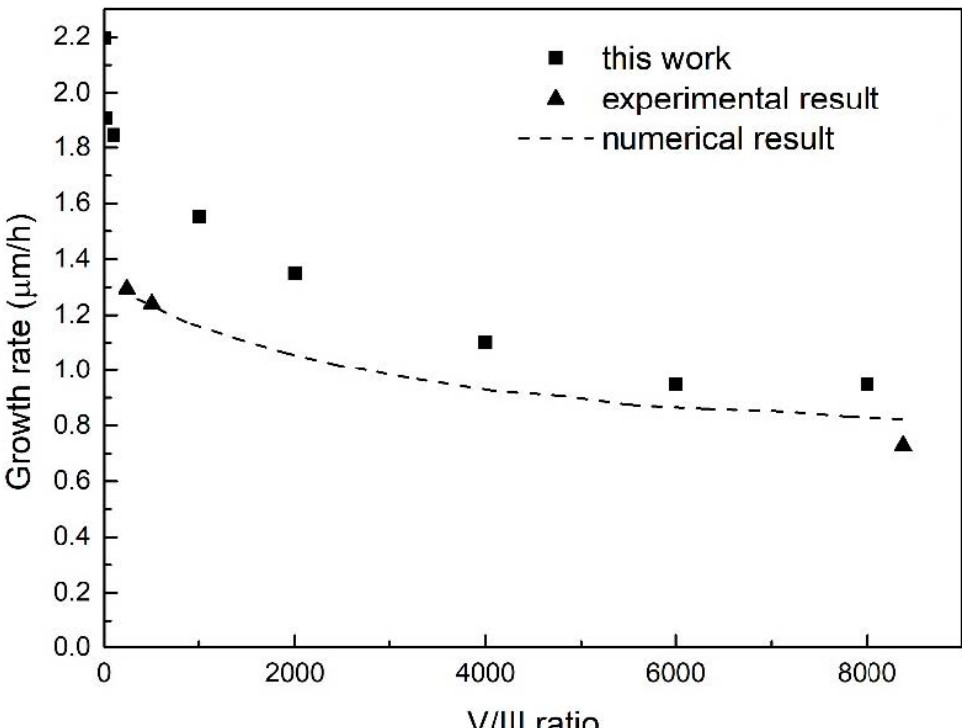

**Figure 2.** Relationship between the average growth rate and the V/III along the entire wafer surface. The square points represent the results obtained in this work and the dashed line and triangles indicate the results from a previous numerical and experimental study [14].

The pulse process considered here first starts with a III pulse, followed by a V pulse, with no purge between the III and V pulses, after which the entire process is cycled repeatedly. Figure 3 shows the average growth rate along the wafer surface during the pulsed process and Figures 4–6 show the mass fraction distributions of TMAl and TMAlNH$_3$ at t = 0.18 s, 0.35 s, and 0.46 s for different V/III ratios. The III pulse reaches the hot substrate surface at approximately t = 0.05 s and the AlN film grows through the pyrolysis reaction [13]. Therefore, there is significant increase in the average growth rate around t = 0.05 s. The V pulse is injected into the chamber at 0.15 s and then the adductive reaction occurs at the rear interface of the III pulse. The adductive reaction between the TMAl and NH$_3$ leads to the formation of Al particles. Figure 4 shows that at t = 0.18 s, the front interface of TMAlNH3 is still noted to reach the region close to the hot substrate. The movement of the front interface of TMAlNH$_3$ is faster with a higher V/III ratio. Due to the stronger adductive reaction, more TMAlNH$_3$ is produced at a higher V/III ratio (Figure 4b). It affects the substrate at approximately t = 0.2 s. In Figure 3, the beginning of an abrupt drop at around t = 0.2 s can be observed, which occurs earlier and faster with a higher V/III ratio.

The second III pulse starts at 0.3 s and the growth rate increases again at approximately t = 0.36 s (Figure 3). However, as the V/III ratio increases, the magnitude of the maximum growth rate becomes smaller. As can be seen from Figure 5a, for higher V/III ratios, the movement of the front interface of the III pulse is slower because more NH$_3$ remains in the chamber from the previous cycle which reacts with the TMAl to form TMAlNH$_3$. The production of TMAlNH3 increases when the V/III ratio is higher (Figure 5b). This results in a greater V/III ratio, and a later and slower increase in the growth rate after t = 0.35 s (Figure 3). Figure 6 shows the mass fraction distributions of TMAl and TMAlNH3 at t = 0.46 s for different V/III ratios, which is 0.01 s after the start of the second V pulse. When the V/III ratio is higher, the rear interface of the III pulse moves faster (Figure 6a), similar to the movement of the rear interface shown in Figure 4a. However, the front interface of the III pulse is affected by the remaining NH$_3$ in the chamber. The larger

the V/III ratio, the greater the formation of $TMAlNH_3$ near the front and rear interfaces (Figure 6b). This makes the region occupied by the TMAl smaller. Therefore, the maximum value of the average growth rate for the second cycle decreases as the V/III ratio increases (Figure 3). Figure 7 shows the mass fraction distribution of TMAl at t = 0.49 s for different V/III ratios. The movement of the rear interface over the wafer with a V/III ratio equal to 10000 can be seen. This results in a rapid drop in the average growth rate because the rear interface causes a rapid drop in TMAl above the wafer. The phenomena in the third and subsequent cycles are similar to those in the second cycle. It is clear that the decrease in the average growth rate for higher V/III ratios is due to greater $TMAlNH_3$ formation. The insertion of a pure $H_2$ pulse between the III and V pulses prevents the $NH_3$ from reacting with the TMAl to form $TMAlNH_3$.

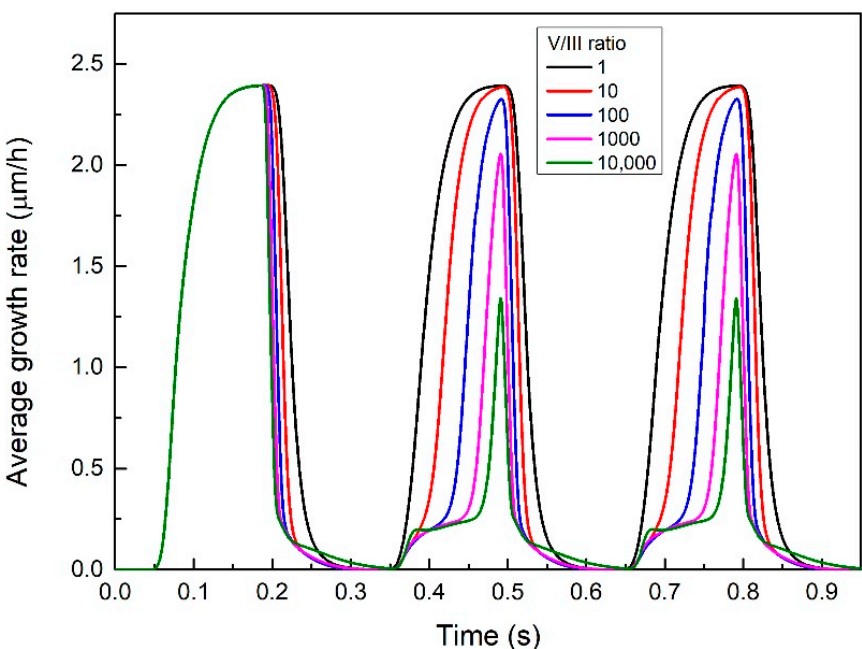

**Figure 3.** Average growth rate along the wafer surface during the pulsed process.

According to our previous results [11], the formation of $TMAlNH_3$ can be minimized by inserting a pure $H_2$ purge pulse between the III and V pulses. On the other hand, there is a decline in the overall growth rate due to a wider cycle time. As can been seen in Figure 3, the growth rate after the first cycle is significantly affected by changes in the V/III ratios. In practice, the pulsed process time is usually greater than several minutes. Therefore, the differential impact of the first cycle on the overall growth rate will be very small. To investigate the effect of the purge time on the overall growth rate, the average growth rate for the second cycle is calculated. Figure 8 shows the average growth rate throughout the second cycle for different purge times and different V/III ratios. The decrease in the average growth rate as the V/III ratio increases occurs because of the formation of TMAlNH3. The adductive reaction is weak when the V/III ratio is equal to 1. The benefit of reducing the formation of $TMAlNH_3$ is outweighed by the detrimental effect of the increasing cycle time which causes the average growth rate to decrease with an increase in purge time. When the V/III ratio is equal to 100, there is an increase in the average growth rate as the purge time increases from 0 s to 0.05 s, because the benefit of the reduction in $TMAlNH_3$ outweighs the disadvantage of the increasing cycle time. However, there is a decrease in the average growth rate when the purge time increases from 0.05 s to 0.1 s. In this case, the generation of particles (formation of $TMAlNH_3$) is very low. The benefit of reducing particle generation is outweighed by the disadvantage of an increase in the cycle time. The maximum growth rate is about 0.75 μm/h. When the V/III ratio is equal to 10,000, the average growth rate increases as the purge time increases. However, the average growth rate may not increase

with longer purge times, because the growth rate at higher V/III ratios does not exceed that obtained at lower V/III ratios. The maximum average growth rate can be expected to be around 0.7 μm/h.

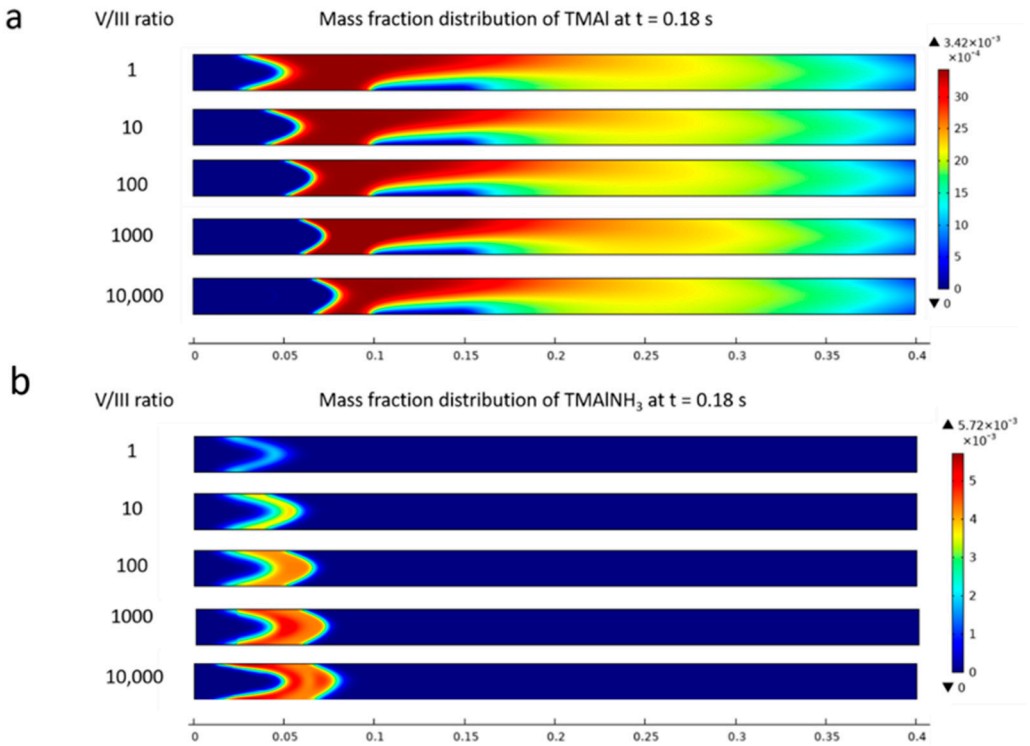

**Figure 4.** Mass fraction distributions of (**a**) TMAl and (**b**) TMAlNH$_3$ at 0.18 s for different V/III ratios.

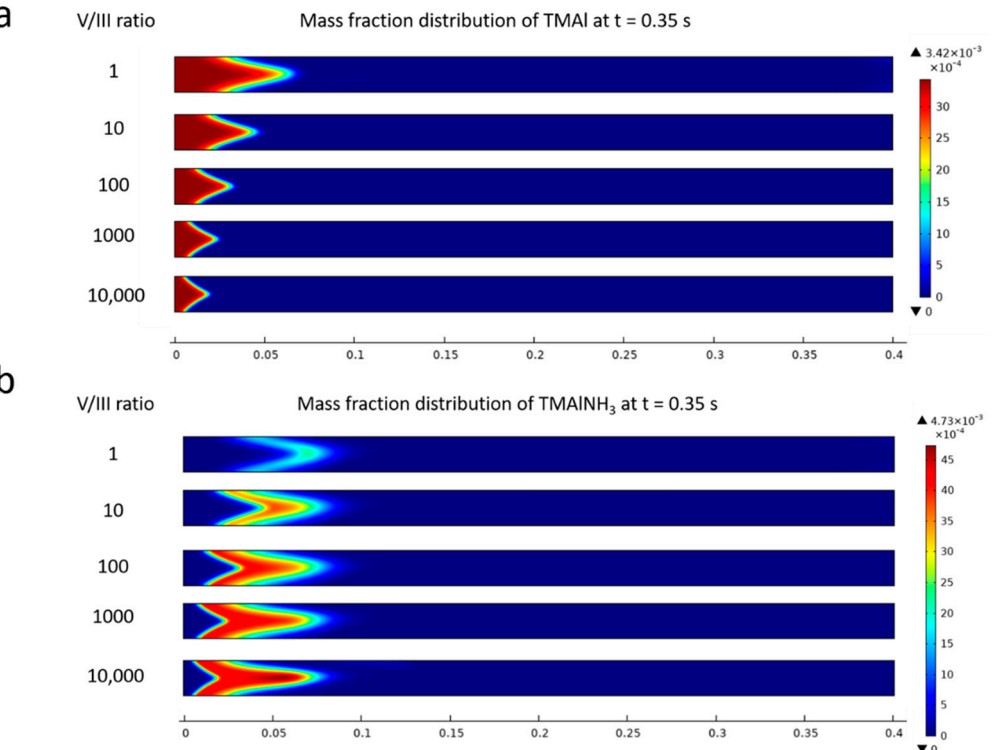

**Figure 5.** Mass fraction distributions of (**a**) TMAl and (**b**) TMAlNH$_3$ at 0.35 s for different V/III ratios.

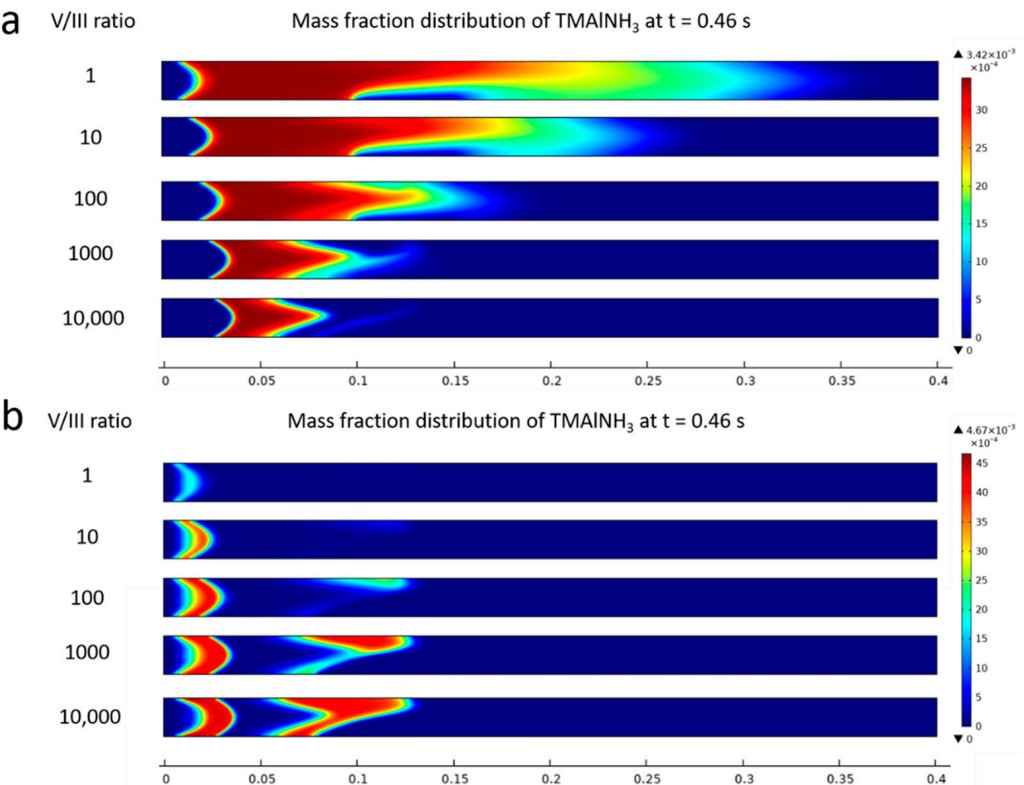

**Figure 6.** Mass fraction distributions of (**a**) TMAl and (**b**) TMAlNH$_3$ at 0.46 s for different V/III ratios.

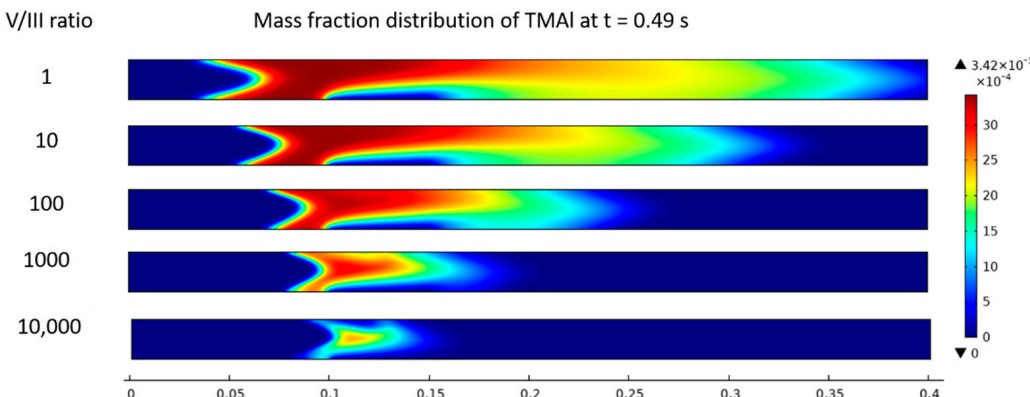

**Figure 7.** Mass fraction distributions of TMAl at 0.49 s for different V/III ratios.

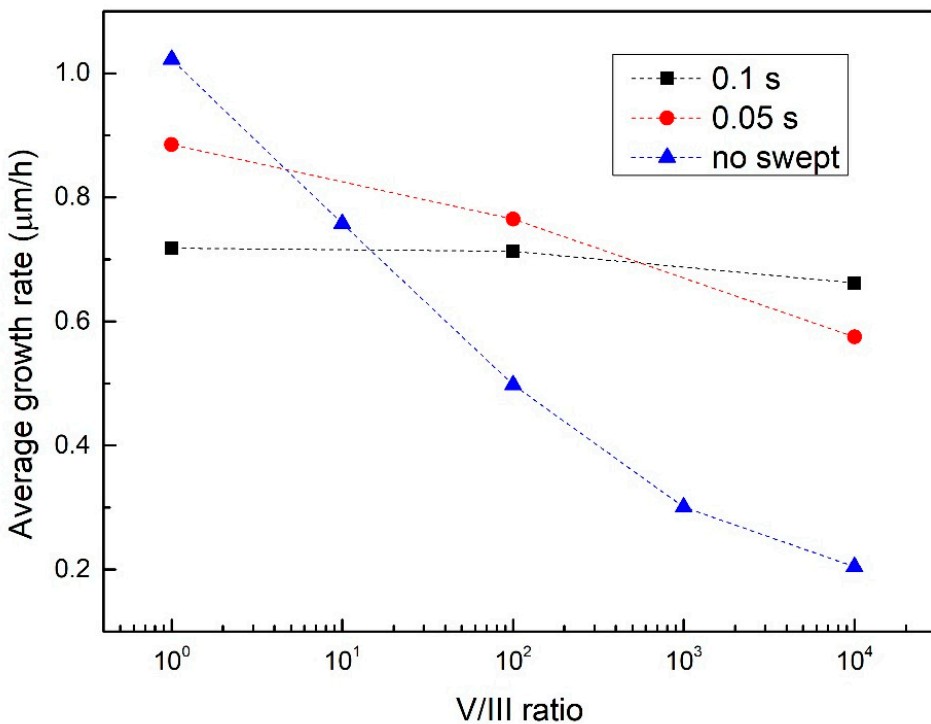

**Figure 8.** Average growth rate in the second cycle for different purge times and V/III ratios.

## 4. Conclusions

A numerical analysis is performed to investigate the effects of the V/III ratio on the growth rate during the pulse injection process. The evolution of TMAl and TMAlNH$_3$ during the process is studied in detail. The adductive reaction can be reduced by inserting a pure H$_2$ purge pulse between III and V pulses, but there may be a decrease in the overall growth due to the longer cycle time. The trade-off between the benefits of reducing particle generation and the disadvantages of increasing the process time can be understood by investigating the relationship between the purge time, the III/V ratio, and the overall growth rate. When the V/III ratio is increased, there is a significant decrease in the average growth rate per cycle, when there is no purge pulse between the III and V pulses due to the stronger adductive reaction (the higher Al particle generation). For smaller V/III ratios, there is a decrease in the growth rate as the purge time increases because the benefit of reducing the adductive reaction is offset by the detrimental effect of the increasing cycle time. This causes the average growth rate to decrease as the purge time increases. The benefit of reducing the adductive reaction is more significant for higher V/III ratios. With increasing purge times, the decline in the growth rate becomes smaller as the V/III ratio increases. Clearly, inserting a pure H$_2$ purge pulse between the III and V pulses can significantly improve the growth rate when the V/III ratios are high. Higher overall growth rates can be obtained at higher V/III ratios by selecting an appropriate purge time.

**Author Contributions:** W.-J.L., conceptualization, methodology, formal analysis, writing—original draft; J.-C.C., conceptualization, writing—review and editing, supervision. All authors have read and agreed to the published version of the manuscript.

**Funding:** The authors would like to thank the Ministry of Science and Technology, R.O.C, for their support of this study through grant number MOST-106-2221-E-008-052-MY3.

**Institutional Review Board Statement:** Not applicable.

**Informed Consent Statement:** Not applicable.

**Data Availability Statement:** Not applicable.

**Conflicts of Interest:** The authors declare no conflict of interest.

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
