# Peer review of "Numerical Study of Growth Rate and Purge Time in the AlN Pulsed MOCVD Process"

_crystals, doi:10.3390/cryst12081101_

Round 1
Reviewer 1 Report
In my opinion the paper does not need any improvements. As it is important for growth of AlN layers, the paper is recommended for publication in Crystals in the present form.
Author Response
Thank you so much for recommending this paper for publication in Crystals.
Reviewer 2 Report
The Paper “Numerical study of the growth rate and purge time in the AlN pulsed MOCVD process” reports on the simulation of MOCVD AlN growth and found some relations between the purge time, growth rate, V-III ratio, etc. The paper is well written and the results are sound. However, there is a lack of physics explanation in many cases. For example, in the introduction, the authors state that a problem is the low migration coefficient of the Al atoms. But how does the pulsed MOCVD process boost the Al migration? What is the mechanism? Why does the pulsed process increase the quality? Why is the higher V-III ratio leading to a lower growth rate? Why should we consider pulsed MOCVD in the first place? There can be even more questions related to the physical mechanism of the growth. The authors are expected to dig into the physics here and give the readers a clear picture, instead of just showing the calculation results. Also, the English should be improved. For example, in the first paragraph of “Results and discussion”, “there is a decrease in the average growth rate decreases due to the increased formation of Al particles“ is a weird sentence.
Author Response
Thanks for reviewer’s suggestion. There are two ways to improve the migration of Al atoms on the reactive surface. The first is to increase the energy of Al atoms on the surface by applying higher growth temperatures. Another approach is to reduce the number of N atoms on the surface, because N atoms interrupt the migration of Al atoms. During the pulsed process, the supply of TMAl and NH3 is carried out separately. The TMAl follows the pyrolysis reaction pathway instead of adducting with NH3 in the gas phase before being deposited on the surface. It makes the dominant species on the surface from Al-N to Al. The migration of Al atom increases because the number of N atoms is much lower during the pulsed process [2]. Due to the lateral growth of film caused by the higher Al migration, smoother AlN film can be achieved. On the other hand, the adductive reaction is more intense during the growth in higher V/III ratios because the concentration of NH3 is higher. It converts more TMAl source into AlN particle that cannot be adsorbed by the surface and cannot become AlN film. In this situation, the growth rate of AlN film is reduced due to the lack of TMAl source. Some sentences have been improved in the revise version.
Round 2
Reviewer 2 Report
It is ok to accept in the present form.